# Conserving symmetries in Bose-Einstein condensate dynamics requires many-body theory

Kaspar Sakmann[1*], Jörg Schmiedmayer[1]

**1 Technische Universität Wien, Vienna Center for Quantum Technologies, Atominstitut, Stadionallee 2, 1020 Vienna, Austria**
*kaspar.sakmann@gmail.com

December 14, 2018

## Abstract

We explain from first principles why satisfying conservation laws in Bose Einstein condensate dynamics requires many-body theory. For the Gross-Pitaevskii mean-field we show analytically and numerically that conservation laws are violated. We provide examples for angular momentum and linear momentum conservation. Arbitrarily large violations occur despite negligible depletion and interaction energy. For the case of angular momentum we show through extensive many-body simulations how the conservation law can be gradually restored on the many-body level. Implications are discussed.

# 1   Introduction

The many-body Schrödinger equation describes the microscopic dynamics of Bose-Einstein condensates (BECs). Conservation laws follow from invariances of the Hamiltonian under symmetry transformations. For instance, conservation of angular momentum is directly linked to invariance of the Hamiltonian under rotations and conservation of momentum to invariance of the Hamiltonian under translations. For a purely distance dependent interparticle interaction and rotationally symmetric trapping potentials, the Hamiltonian and the angular momentum operator commute and the exact dynamics conserves angular momentum for any initial state. Similarly, in the absence of a trapping potential the Hamiltonian is translationally invariant and the total momentum is conserved in the exact dynamics for any initial state.

However, in practice the many-body Schrödinger equation can rarely be solved exactly and an ansatz for the wave function must be made for which equations of motion are derived. The simplest approach is to expand the many-body wave function in a set of permanents constructed from a finite number of time-independent orbitals, e.g. harmonic oscillator eigenfunctions or the solution to some other single-particle problem. This direct diagonalization approach preserves the linearity and the symmetries of the many-body Schrödinger equation. Despite these desirable properties, a direct diagonalization is hardly ever feasible for more than a few particles, because the many-body Hilbert space grows far too quickly with the number of particles and the number of spatial dimensions.

A very successful alternative to direct diagonalization builds on the time-dependent variational principle, put forward by Dirac and Frenkel [1,2], see also [3]. The idea is to include the orbitals themselves in the set of variational parameters and to let the time-dependent variational principle determine their shape. Thereby, the Hilbert space that can be explored by an ansatz wave function is much larger than that of a corresponding expansion in predetermined

orbitals.

The most famous example of this approach is Hartree-Fock theory, developed in the late 1920s for electrons, where the many-body wave function is approximated by a single Slater determinant, determined by the variational principle. Unlike the Schrödinger equation the Hartree-Fock equations are nonlinear. Nowadays it is well-known that their solutions violate conservation laws that are satisfied by the many-body Hamiltonian [4]. However, it took several decades until it was noticed: only in 1963 P. O. Löwdin pointed out *"... the solution D corresponding to the absolute [energy] minimum has now usually lost its eigenvalue property with respect to [the constant of the motion] Λ, i.e., the corresponding Hartree-Fock functions are no longer symmetry-adapted"* [5].

Applying the Hartree-Fock ansatz to bosons leads to the well-known Gross-Pitaevskii (GP) mean-field approximation [6, 7]. Analogous to the fermionic case, stationary GP solutions violate symmetries that are present in the many-body Hamiltonian. For instance, the exact ground state of $N$ attractively interacting bosons in 1D free space is delocalized and satisfies the translational symmetry of the Hamiltonian, but the corresponding GP solution – often called bright soliton solution– is localized and violates it. see Ref. [8]

Over the last half a century, many attempts were undertaken to restore symmetries of symmetry breaking mean-field solutions. A very popular method became known as the Hill-Wheeler integral equation, see e.g. [4,9]. An exhaustive list of its uses in nuclear and molecular physics is beyond the scope of this work. The following references provide symmetry restored approximations to the ground state of Bose-Einstein condensates. In [10] the authors restore the translational symmetry of the symmetry-violating GP ground state. In [11] the authors restore the rotational symmetry for a 2D strongly interacting BEC in a harmonic trap and in [12] the rotational symmetry of symmetry broken GP vortex cluster solutions. Summarizing, not only the fact that stationary GP solutions violate conservation laws, but also the methods to restore such symmetries have been textbook material for nearly forty years [4].

Here, we address exclusively the *time-dependent* problem. Given that the many-body Hamiltonian commutes with an observable $A$ and an initial state, under what circumstances is $A$ conserved in the time-dynamics of a variational approximation? To answer this question we analyze the properties of the equations of motion that follow from the time-dependent variational principle. We use a many-body ansatz that includes the popular GP mean-field as a special case. This ansatz is known as the multiconfigurational time-dependent Hartree for bosons (MCTDHB) ansatz [13, 14]. The MCTDHB ansatz wave function includes all permanents that can be constructed by distributing $N$ bosons over $M$ orbitals. For $M \to \infty$ the MCTDHB method converges to the exact solution of the many-body Schrödinger equation; for the same accuracy MCTDHB requires much fewer orbitals than an expansion in time-independent orbitals, see e.g. Refs. [15, 16] for details.

Using the MCTDHB method it was found previously that the center of mass coordinate and total momentum operators provide sensitive probes of many-body correlations, e.g. through their variances [17,18] and full counting distributions [19], even if the condensate is only weakly interacting. Here, we are following this line of research.

We provide a first principle explanation *why* observables that commute with the Hamiltonian can only be conserved on the many-body level. As a measure for the amount of the violation we track the time evolution of the second moment of the observable that commutes with the Hamiltonian. The main example we provide is conservation of angular momentum for a BEC that is initially displaced from the center of a harmonic trap and is allowed to slosh back and forth. We show analytically that the rate of the violation of angular momentum

conservation is proportional to the GP interaction parameter, the initial momentum and the initial displacement.

In the corresponding numerical simulations we use an interaction strength that is so small, that the depletion and the interaction energy are practically negligible. Thus, one would expect no effects beyond GP mean-field to be of noticeable magnitude. We show that this is not the case. Conserving angular momentum in the dynamics requires a far greater Hilbert space than that spanned by the GP mean-field ansatz, despite the very favorable parameter values.

Finally, we address the free space case and the associated conservation of total momentum. In an analytic example we show that the Gross-Pitaevskii mean-field dynamics also violates the conservation of linear momentum. We provide the parametric dependence of this violation.

## 2  General Theory

### 2.1  Many-body Schrödinger equation

We use units where $\hbar = m = 1$ and all spatial distances are measured in multiples of a length scale $L$. Formally, we divide the dimensional many-body Hamiltonian by $\hbar^2/(mL^2)$, where $m$ is the mass of a boson. The time-dependent many-body Schrödinger equation then reads

$$i\frac{\partial \Psi}{\partial t} = H\Psi \tag{1}$$

and

$$H = \sum_{j=1}^{N} h(\mathbf{r}_j) + \sum_{j<k} W(\mathbf{r}_j - \mathbf{r}_k). \tag{2}$$

Here, $\Psi = \Psi(\mathbf{r}_1, \ldots, \mathbf{r}_N, t)$ is the many-body wave function, $h(\mathbf{r}) = -\frac{\Delta}{2} + V(\mathbf{r})$ is the one-body part of the Hamiltonian, $V(\mathbf{r})$ an external potential and $W(\mathbf{r})$ is a two-body interaction potential.

Given a complete set of orthonormal orbitals $\{\phi_j\}_{j=1,2\ldots,\infty}$ the bosonic field operator is expanded as $\hat{\Psi}(\mathbf{r}) = \sum_i b_i \phi_i(\mathbf{r})$ with $[\mathbf{\Psi}(\mathbf{r}), \mathbf{\Psi}^\dagger(\mathbf{r}')] = \delta(\mathbf{r}-\mathbf{r}')$. The operators $b_j = \int d\mathbf{r} \phi_j(\mathbf{r})\mathbf{\Psi}(\mathbf{r})$ annihilate a boson in the orbital $\phi_j(\mathbf{r})$ and satisfy $[b_i, b_j^\dagger] = \delta_{ij}$.

### 2.2  Reduced-density matrices, Bose-Einstein condensation, Fragmentation

Reduced density matrices are central to the theory of Bose-Einstein condensates [20, 21]. Here and in the following, we suppress the time argument, whenever no ambiguity arises. The one- and two-particle reduced density matrices are defined as

$$
\begin{aligned}
\rho^{(1)}(\mathbf{r}|\mathbf{r}') &= \langle\Psi|\mathbf{\Psi}^\dagger(\mathbf{r}')\mathbf{\Psi}(\mathbf{r})|\Psi\rangle \\
&= \sum_{i,j} \rho_{ij}\phi_i^*(\mathbf{r}')\phi_j(\mathbf{r}) \\
\rho^{(2)}(\mathbf{r}_1, \mathbf{r}_2|\mathbf{r}_1', \mathbf{r}_2') &= \langle\Psi|\mathbf{\Psi}^\dagger(\mathbf{r}_1')\mathbf{\Psi}^\dagger(\mathbf{r}_2')\mathbf{\Psi}(\mathbf{r}_2)\mathbf{\Psi}(\mathbf{r}_1)|\Psi\rangle \\
&= \sum_{i,j,k,l} \rho_{ijkl}\phi_i^*(\mathbf{r}_1')\phi_j^*(\mathbf{r}_2')\phi_k(\mathbf{r}_1)\phi_j(\mathbf{r}_2),
\end{aligned}
\tag{3}
$$

respectively and contain all information about one- and two-particle correlations. Their matrix elements are given by

$$
\begin{aligned}
\rho_{ij} &= \langle\Psi|b_i^\dagger b_j|\Psi\rangle \\
\rho_{ijkl} &= \langle\Psi|b_i^\dagger b_j^\dagger b_k b_l|\Psi\rangle.
\end{aligned}
\tag{4}
$$

with $\sum_i \rho_{ii} = N$ and $\sum_i \rho_{iiii} = N(N-1)$. By diagonalizing the one-particle RDM one obtains

$$
\rho^{(1)}(\mathbf{r}|\mathbf{r}') = \sum_i n_i^{(1)}\alpha_i^*(\mathbf{r}')\alpha_i(\mathbf{r})
\tag{5}
$$

where the eigenfunctions $\alpha_i(\mathbf{r})$ and eigenvalues $n_i^{(1)}$ are known as natural orbitals and natural occupations and satisfy $\int d\mathbf{r}' \rho^{(1)}(\mathbf{r}|\mathbf{r}')\alpha_i(\mathbf{r}_i') = n_i^{(1)}\alpha_i(\mathbf{r})$. It is common to order the eigenvalues decreasingly $n_1^{(1)} \geq n_1^{(2)} \geq \ldots$ and so on. Bose-Einstein condesation is defined as follows. If one and only one eigenvalue $n_i^{(1)} = \mathcal{O}(N)$ exists, the system of bosons is said to be condensed [22]. If more than one such eigenvalue exists, the system of bosons is said to be fragmented [23, 24]. In a pure condensate only one orbital is occupied and $n_1^{(1)} = N$ and all other natural occupations are zero. The case of a depleted condensate is obtained when $n_1^{(1)} \approx N$ and all other occupations are much smaller than $N$. In this work we only study depleted condensates.

## 2.3 Conservation of observables

In the following $A$ denotes an observable defined as

$$
A = \sum_{j=1}^{N} a_j = \sum_{ij} b_i^\dagger b_j \langle\phi_i|a|\phi_j\rangle
\tag{6}
$$

where $a$ is a hermitian single-particle operator, e.g. $a = p$. As is well known, an observable $A$ is conserved, if $[H, A] = 0$. An immediate consequence is that all moments of $A$ are constant in time. This can be seen as follows. Using $[H, A] = 0$ and

$$
[H, A^n] = nA^{n-1}[H, A] = 0
\tag{7}
$$

as well as the Baker-Campbell-Hausdorff identity

$$
e^F G e^{-F} = G + [F, G] + \frac{1}{2!}[F[F, G]] + \ldots
\tag{8}
$$

one finds $\langle A^n\rangle(t) = \langle\Psi(0)|e^{iHt}A^n e^{-iHt}|\Psi(0)\rangle = \langle\Psi(0)|A^n|\Psi(0)\rangle$ and thus

$$
\frac{d}{dt}\langle A^n\rangle(t) = 0,
\tag{9}
$$

for *any* initial state $\Psi(0)$. This implies that in order to fulfill a conservation law not only the expectation value $\langle A\rangle$, but also all higher moments $\langle A^n, n > 1$ must be constant in time. Another proof of (9) can be found in Appendix A. Note that Eq. (9) holds for any quantum system, regardless of the number of particles or the interaction strength.

The operator $A$ in Eq. (6) is a one-body operator, i.e. it only involves the one-particle reduced density matrix. Using (4) its expectation value is given by

$$\langle A \rangle = \sum_{ij} \rho_{ij} \langle \phi_i | a | \phi_j \rangle. \tag{10}$$

$A^n$ on the other hand acts on up to $n$-particles and thus reduced density matrices up to order $n \leq N$ are involved. Specifically for the second power:

$$A^2 = \sum_{j=1}^{N} a_j^2 + \sum_{i \neq j}^{N} a_i a_j \tag{11}$$

$$\tag{12}$$

i.e. $A^2$ is a sum of one- and two-body operators. Using (4) the expectation value $\langle A^2 \rangle$ can be expressed as

$$\langle A^2 \rangle = \sum_{i,j} \rho_{ij} \langle \phi_i | a^2 | \phi_j \rangle + \sum_{ijkl} \rho_{ijkl} \langle \phi_i | a | \phi_k \rangle \langle \phi_j | a | \phi_l \rangle \tag{13}$$

In this work we focus on the first and second moments of $A$. In Appendix B we discuss analogous results for the third moment, but the results presented below only require the first two moments of $A$.

We would like to add an experimental perspective to the above. In BEC experiments it is often the case that an ensemble average is implicitly performed by simultaneously measuring the outcome of a large number of more or less similar copies of an experimental setup. Such experiments measure the average value of an observable and thus cannot test whether $\frac{d}{dt} \langle A^n \rangle = 0$ is satisfied for $n > 1$. However, more recently a number of experiments have been carried out that measure full distribution functions [25–28]. Such experiments are well-suited for measuring the time-independence of the moments $\langle A^n \rangle$, as required by Eq. (9) for a conserved observable.

## 2.4   Distribution function of an observable

Let us expand on what Eq. (9) means for an experiment in which $A$ is measured. We write $\mathcal{A}$ for the outcome of a measurement of $A$. If the experiment is repeated many times the measurements $\{\mathcal{A}_i\}_{i=1,2,\dots}$ are distributed according to

$$\mathcal{A} \sim P(\mathcal{A}). \tag{14}$$

Mathematically, Eq. (9) then expresses the fact that $P(\mathcal{A})$ of a conserved observable $A$ is stationary. This can be rigorously proven assuming sufficient regularity of the moments: in this case $P(\mathcal{A})$ has an analytic characteristic function

$$\chi(q) = \langle e^{iqA} \rangle = \sum_{n=0}^{\infty} \frac{(iq)^n}{n!} \langle A^n \rangle \tag{15}$$

and can be uniquely reconstructed from its moments through

$$P(\mathcal{A}) = \frac{1}{2\pi} \int_{-\infty}^{\infty} dq \, e^{-iq\mathcal{A}} \chi(q). \tag{16}$$

Now consider the case where $A$ has a discrete spectrum $A\Psi_j = \mathcal{A}_j \Psi_j$ and is conserved, $[H, A] = 0$. Expanding the wave function $\Psi(t) = \sum_j c_j(0)e^{-iE_j t}\Psi_j$ in the eigenfunctions common to $A$ and $H$ one obtains

$$\langle A^n \rangle(t) = \sum_j |c_j(t)|^2 \mathcal{A}_j^n \tag{17}$$

and substituting into (16) one arrives at

$$P(\mathcal{A}) = \sum_j |c_j(t)|^2 \delta(\mathcal{A} - \mathcal{A}_j) = \sum_j |c_j(0)|^2 \delta(\mathcal{A} - \mathcal{A}_j) \tag{18}$$

which is time-independent. Thus, conserved observables have stationary probability distributions $P(\mathcal{A})$. For details on the *moment problem*, see e.g. [29]. For its application to quantum mechanics, see e.g. [30].

## 2.5 Gedankenexperiment

As an instructive example consider a *Gedankenexperiment* in which a single particle is prepared in an initial state $\phi$ and after some time either momentum or position are measured. We assume that the particle propagates in free space in one dimension, i.e. the Hamiltonian is $H = \frac{p^2}{2}$ and $\mathbf{r} = x$. We write $\tilde{\phi}(p)$ for the wave function in momentum space. Clearly, $[H, p] = 0$, so momentum is conserved. To see that Eq. (9) is satisfied we define $E_p = \frac{p^2}{2}$ and $\tilde{\phi}(p) = \tilde{\phi}(p, t = 0)$. Then using

$$\phi(x, t) = \frac{1}{2\pi} \int dp \tilde{\phi}(p, t) e^{ipx} \tag{19}$$

one finds

$$\begin{aligned}
\langle p^n \rangle(t) &= \int dx \phi(x, t) \left(-i\frac{d}{dx}\right)^n \phi(x, t) \\
&= \iiint \frac{1}{(2\pi)^2} dp' dp dx \tilde{\phi}^*(p') e^{-i(p'x - E_p' t)} p^n \tilde{\phi}(p) e^{i(px - E_p t)} \\
&= \int dp dp' \frac{1}{2\pi} \delta(p - p') |\phi(p)|^2 p^n \\
&= \int dp \frac{1}{2\pi} |\phi(p)|^2 p^n \\
&= \langle p^n \rangle(0)
\end{aligned} \tag{20}$$

and hence Eq. (9) is fulfilled. Let us now specify the shape of the wave packet. We consider a particle at rest with a wave function given by a Gaussian

$$\phi(x, 0) = \left(\frac{1}{\pi \sigma_0^2}\right)^{1/4} e^{-\frac{x^2}{2\sigma_0^2}}. \tag{21}$$

Propagating the Schrödinger equation for the wave packet (21) one finds

$$\langle x^2 \rangle(t) = \frac{\sigma_0^2}{2} \left(1 + \left(\frac{t}{\sigma_0^2}\right)^2\right) \tag{22}$$

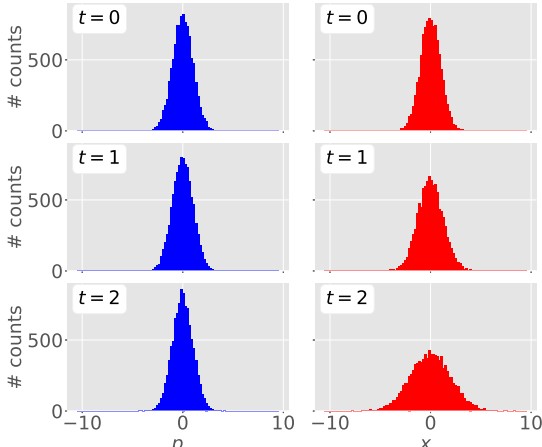

Figure 1: **Gedankenexperiment.** Shown are $10^4$ samples of the exact momentum (left) and position (right) distributions for a single particle in one-dimensional free space at different times. Momentum is conserved and accordingly the momentum distribution is stationary. However, the position operator does not commute with the Hamiltonian and the position distribution is time-dependent. The initial wave function is a Gaussian of width $\sigma_0 = \sqrt{2}$. See text for details. All quantities shown are dimensionless.

which spreads during time evolution. This is completely expected, because $[H, x] = -ip \neq 0$, i.e. $x$ is not conserved.

Fig. 1 shows random samples from the exact position and momentum distributions during the time evolution of the wave packet (21) with initial width $\sigma_0 = \sqrt{2}$. The time-dependence is clearly visible in the position distribution, while the momentum distribution is stationary. An experimenter would correctly conclude that momentum is conserved, but position is not. Note that the expectation value of the position operator, $\langle x \rangle = 0$, is time-independent, even though $x$ is not conserved $[H, x] \neq 0$ in this example. This example nicely demonstrates that it cannot be concluded that an observable $A$ is conserved, based merely on $\frac{d}{dt}\langle A \rangle = 0$.

# 3    The time-dependent variational principle

The theory developed so far is exact. However, it only applies to exact analytical solutions of the many-body Schrödinger equation and only few analytical solutions are known. In order to describe realistic quantum many-body systems approximations are inevitable and in practical computations an ansatz for the many-body wave function using a finite number of basis vectors needs to be made.

## 3.1    Dirac-Frenkel formulation

The Dirac-Frenkel variational principle, analogous to d'Alembert's principle in classical mechanics, states that for a given parametrized ansatz wave function $\Psi$ the dynamics is to be

determined by the condition [1–3]

$$\langle\delta\Psi|(H - i\frac{\partial}{\partial t})|\Psi\rangle = 0. \tag{23}$$

In (23), $\delta\Psi$ denotes any variation that lies in the space $\{\delta\Psi\}$ of linear variations of $\Psi$. In words the principle states that the residual $(H - i\frac{\partial}{\partial t})|\Psi\rangle$ is orthogonal to the space of allowed variations. The space of allowed variations $\{\delta\Psi\}$ is assumed to be linear complex here, i.e. if $\delta\Psi$ is an allowed variation, then $i\delta\Psi$ is also an allowed variation [1, 3].

There are two other commonly employed variational principles, namely the McLachlan variational principle [31], and a Lagrangian formulation, where stationarity of the action integral is required, also known as the principle of least action [3]. These two alternative variational principles amount to restricting (23) to its imaginary or real part, respectively [32]. If the space $\{\delta\Psi\}$ is linear complex all three variational principles are equivalent [32]. The complex linear case is the most common and the Dirac-Frenkel formulation is then preferable due to its simplicity. For ansatz wave functions with only real linear spaces $\{\delta\Psi\}$, e.g. ansatz wave functions that depend on purely real parameters, the different variational principles are not equivalent and lead to different equations of motion [32].

## 3.2   Conservation laws and the variational principle

First we review an important result about conservation laws satisfied by ansatz wave functions that follow equations of motion determined by the Dirac-Frenkel variational principle [33, 34].

**Theorem 1.** Let the self-adjoint operator $A$ commute with the Hamiltonian $H$, i.e. $[H, A] = 0$. If $A\Psi$ lies in the space of the allowed variations $\{\delta\Psi\}$ of the ansatz wave function $\Psi$ then $\langle A\rangle$ is time-independent, i.e.

$$\frac{d}{dt}\langle A\rangle = 0. \tag{24}$$

*Proof.*

$$\begin{aligned}
\frac{d}{dt}\langle A\rangle &= 2\,\mathrm{Re}\langle A\Psi\frac{\partial}{\partial t}|\Psi\rangle \\
&= -2\,\mathrm{Im}\langle A\Psi|H|\Psi\rangle \\
&= 2\langle\Psi|[H, A]|\Psi\rangle = 0, \tag{25}
\end{aligned}$$

using (23) and $A\Psi \in \{\delta\Psi\}$ in the second equality. $\qquad\square$

The result demonstrates the importance of the space $\{\delta\Psi\}$. In general $A\Psi$ does not lie entirely in $\{\delta\Psi\}$, depending on the ansatz wave function and the operator $A$. However, the larger $\{\delta\Psi\}$ is, the smaller the error will be. Conversely, if $[H, A] = 0$ and $A\Psi$ lies in the space $\{\delta\Psi\}$, the equations of motion derived from the Dirac-Frenkel variational principle (23) are guaranteed to keep the first moment $\langle A\rangle$ constant at all times. In the following we will investigate the structure of the space $\{\delta\Psi\}$ for different ansatz wave functions.

# 4 Multiconfigurational Time-Dependent Hartree for bosons

## 4.1 Equations of motion

A particularly successful application of the time-dependent variational principle for ultracold bosons is the multiconfigurational time-dependent Hartree for bosons (MCTDHB) method [13, 14]. As opposed to the usual approach of expanding a many-body wave function in a set of fixed orbitals, the orbitals in the MCTDHB method are allowed to depend on time and are included in the set of variational parameters of the ansatz wave function. We briefly summarize the MCTDHB method. To this end we consider $M$ time-dependent orbitals $\{\phi_j(\mathbf{r}, t)\}, j = 1, \ldots, M$. The MCTDHB ansatz for the many-boson wave function then reads

$$|\Psi(t)\rangle = \sum_{\vec{n}} C_{\vec{n}}(t)|\vec{n}; t\rangle, \tag{26}$$

where the sum over $\vec{n} = (n_1, \ldots, n_M)^T$ runs over all $\binom{N+M-1}{N}$ permanents

$$|\vec{n}; t\rangle = \frac{b_1^{\dagger n_1}(t) b_2^{\dagger n_2}(t) \ldots b_m^{\dagger n_m}(t)}{\sqrt{n_1! n_2! \ldots n_M!}} |vac\rangle \tag{27}$$

that can be constructed by distributing $N$ bosons over the $M$ orbitals $\{\phi_j(x, t)\}$. The $\{C_{\vec{n}}(t)\}$ are time-dependent expansion coefficients. The equations of motion for the coefficients $\{C_{\vec{n}}(t)\}$ and the orbitals $\{\phi_j(x, t)\}$ can be derived from the Dirac-Frenkel variational principle (23), see [14] for the derivation, which is lengthy but otherwise straightforward. The final result is a coupled system of equations and reads:

$$i\frac{\partial}{\partial t}|\phi_j\rangle = \mathbf{P}\left[h|\phi_j\rangle + \lambda_0 \sum_{k,s,l,q=1}^{M} \{\boldsymbol{\rho}(t)\}_{jk}^{-1} \rho_{kslq}\phi_s^*\phi_l|\phi_q\rangle\right], \tag{28}$$

with $j = 1, \ldots, M$ for the time-dependent orbitals and

$$\mathbf{H}(t)\mathbf{C}(t) = i\frac{\partial}{\partial t}\mathbf{C}(t), \qquad H_{\vec{n}\vec{n}'}(t) = \langle \vec{n}; t|H|\vec{n}'; t\rangle \tag{29}$$

for the expansion coefficients. Here, $\mathbf{P} = 1 - \sum_{u=1}^{M} |\phi_u\rangle\langle\phi_u|$ is a projector on the complementary space of the time-dependent orbitals, $\boldsymbol{\rho}(t)$ is the reduced one-particle density matrix with elements $\rho_{ij}$, $\rho_{kslq}$ are the elements of the two-particle reduced density matrix and $\mathbf{C}(t) = \{C_{\vec{n}}(t)\}$ is the vector of expansion coefficients.

The use of $M$ optimized time-dependent orbitals leads to much faster numerical convergence to the full many-body Schrödinger results than an expansion in $M$ time-independent orbitals. Thereby, problems involving large numbers of bosons can be solved on the full-many-body Schrödinger level. For instance converged results in closed [19, 35] and open [36] trap potentials have been obtained and the method has been favorably benchmarked against an exactly-solvable many-boson model [16]. In practice, Eq. (29) is simplified by a mapping of the configuration space which enables the efficient handling of millions of time-adaptive permanents [37].

## 4.2 Space of allowed variations

The space of allowed variations for the MCTDHB ansatz (26) is obtained by performing variations of the wave function with respect to its coefficients $C_{\vec{n}}(t)$ and its orbitals $\{\phi(\mathbf{r}, t)\}$. The variation with respect to the coefficients leads to

$$\frac{\delta \langle \Psi |}{\delta C_{\vec{n}}^*(t)} = \langle \vec{n}; t|. \tag{30}$$

Thus $\{\delta \Psi\}$ contains all linear combinations of permanents $\langle \vec{n}; t|$. In particular $|\Psi(t)\rangle$ itself lies in $\{\delta \Psi\}$. Conservation of the norm of the wave function thus follows from putting $A = 1$ in Theorem (1).

For variations with respect to the orbitals one can derive the following useful identities [14]. With $\mathbf{\Psi}(\mathbf{r}) = \sum_k \phi_k(\mathbf{r}, t) b_k(t)$ and $b_k(t) = \int \phi_k(\mathbf{r}, t) \mathbf{\Psi}(\mathbf{r})$ it follows that

$$\frac{\delta b_k(t)}{\delta \phi_q^*(\mathbf{r}, t)} = \delta_{kq} \mathbf{\Psi}(\mathbf{r}). \tag{31}$$

Furthermore, using (31) the variation of a permanent is given by

$$\frac{\delta \langle \vec{n}; t|}{\delta \phi_q^*(\mathbf{r}, t)} = \langle \vec{n}; t| b_q^\dagger(t) \mathbf{\Psi}(\mathbf{r}) \tag{32}$$

By summing over the permanents one obtains

$$\frac{\delta \langle \Psi |}{\delta \phi_q^*(\mathbf{r}, t)} = \langle \Psi | \sum_k b_q^\dagger(t) b_k(t) \phi_k(\mathbf{r}, t). \tag{33}$$

## 4.3 Stationarity of the expectation values of one-body observables

We show that the expectation values of one-body observables are stationary. To this end consider the functional

$$F = \langle \Psi | (H - i \frac{\partial}{\partial t}) | \Psi \rangle \tag{34}$$

and the variation $\delta \phi_q^*(\mathbf{r}, t)$. Using (23) and (33) one finds

$$\begin{aligned} \delta F &= \int d\mathbf{r} \frac{\delta F}{\delta \phi_q^*(\mathbf{r}, t)} \delta \phi_q^*(\mathbf{r}, t) \\ &= \langle \Psi | \sum_k a_{qk}(t) b_q^\dagger(t) b_k(t) \left( H - i \frac{\partial}{\partial t} \right) | \Psi \rangle = 0. \end{aligned} \tag{35}$$

with $a_{qk}(t) = \int d\mathbf{r} \delta \phi_q^*(\mathbf{r}, t) \phi_k(\mathbf{r}, t)$. By summing (35) over $q$ we arrive at

$$\langle \Psi | A \left( H - i \frac{\partial}{\partial t} \right) | \Psi \rangle = \langle A\Psi | \left( H - i \frac{\partial}{\partial t} \right) | \Psi \rangle = 0 \tag{36}$$

with the operator $A = \sum_{qk} a_{qk}(t) b_q^\dagger(t) b_k(t)$. Thus for any one-body operator the vector $|A\Psi\rangle$ lies in the space of allowed variations. Formally we can write

$$\{\delta \Psi\} = \text{span}\{|\vec{n}; t\rangle, |A\Psi\rangle\}, \tag{37}$$

i.e. the space of allowed variations is the span of all permanents in the ansatz wave function and all vectors that can be constructed by applying a one-body operator to $\Psi$. If $[H, A] = 0$, it follows from Theorem (1) that $\langle A \rangle$ is time-independent. Thus one-body operators that commute with $H$ are guaranteed to have stationary expectation values by the MCTDHB equations of motion, even if only $M = 1$ orbital is used. Note that vectors of the form $|B\Psi\rangle$, where $B$ is a two-body operator will generally not lie entirely in the space of allowed variations. A very relevant example would be for instance $B = A^2$. For this reason it is important, that $\{\delta\Psi\}$ is a sufficiently large space.

## 4.4   Gross-Pitaevskii mean-field theory

The Gross-Pitaevskii mean-field equation is obtained from the MCTDHB ansatz wave function (26) by setting $M = 1$ and a contact interaction $W(\mathbf{r}) = \lambda_0 \delta(\mathbf{r})$ for the interaction potential. The only contributing permanent is

$$|\vec{n}; t\rangle = |N; t\rangle, \tag{38}$$

where $\langle x | b^\dagger(t) | vac \rangle = \phi(x, t)$. The MCTDHB equation of motion for the orbital (28) then reduces to

$$h_{GP} = h + \lambda |\phi|^2 \tag{39}$$

$$i \frac{\partial}{\partial t} |\phi\rangle = h_{GP} |\phi\rangle, \tag{40}$$

where we used $\rho_{11} = N$, $\rho_{1111} = N(N - 1)$, defined $\lambda = \lambda_0(N - 1)$ and have eliminated the projector $\mathbf{P} = 1 - |\phi\rangle\langle\phi|$ by the global phase transformation $\phi \rightarrow \phi e^{i \int^t \mu(t) dt'}$, with $\mu(t) = \langle\phi|(h + \lambda_0(N - 1)|\phi|^2)|\phi\rangle$. All dynamics in Gross-Pitaevskii theory occurs in the orbital $\phi(\mathbf{r}, t)$.

The space $\{\delta\Psi\}$ in the GP mean-field approximation is the subset of (37) that contains only the state $\Psi$ itself and all vectors that can be contructed by applying a one-body operator to $\Psi$.

$$\{\delta\Psi\} = \text{span}\{|\Psi\rangle, |A\Psi\rangle\}. \tag{41}$$

Therefore, the expectation value $\langle A \rangle$ of any one-body operator $A$, that commutes with the full many-body Hamiltonian is time-independent in the GP dynamics, $\frac{d}{dt}\langle A \rangle = 0$. As illustrated above in $\frac{d}{dt}\langle A \rangle = 0$ does not imply, that $A$ is conserved. A necessary condition for $A$ to be conserved is that *all* moments $\frac{d}{dt}\langle A^n \rangle = 0$ are time-independent, see Eq. (9). To what extent the observable $A$ is conserved in the GP dynamics depends entirely on the extent to which the vectors $A^n\Psi$ are contained in the space (41). This in turn depends strongly on the operator $A$ as well as the current state $\Psi$ and is thus a function of time.

## 5   Mean-Field time-derivative of the moments

We assume $[H, A] = 0$ for a one-body observable $A$ as defined in Eq. (6). Choosing the many-body state (38) the expressions (10) and (13) then reduce to

$$\langle A \rangle_{GP} = N\langle\phi|a|\phi\rangle \tag{42}$$

$$\langle A^2 \rangle_{GP} = N[\langle\phi|a^2|\phi\rangle + (N - 1)\langle\phi|a|\phi\rangle^2]. \tag{43}$$

and by using Eqs. (39-40)

$$
\begin{aligned}
\frac{d}{dt}\langle a^n \rangle &= 2\operatorname{Re}\langle \phi | a^n | \frac{\partial}{\partial t} \phi \rangle \\
&= 2\operatorname{Im}\langle \phi | a^n h_{GP} | \phi \rangle \\
&= -2\lambda \operatorname{Im}\langle \phi || \phi |^2 a^n | \phi \rangle
\end{aligned}
\tag{44}
$$

Taking the time derivative of Eqs. (42-43) and using Theorem 1, leads to

$$
\frac{d}{dt}\langle A \rangle_{GP} = 0
\tag{45}
$$

$$
\frac{d}{dt}\langle A^2 \rangle_{GP} = -2N\lambda \operatorname{Im}\langle \phi || \phi |^2 a^2 | \phi \rangle
\tag{46}
$$

From Eq. (46) it is clear that the GP dynamics violates the conservation of $A$ if the right-hand side is nonzero: $[H, A] = 0$ leads to stationary moments $\frac{d}{dt}\langle A^n \rangle = 0$ and thus to a stationary distribution $P(\mathcal{A})$, see Eqs. (15-16), in contrast to Eq. (46).

An interesting point to note is that the contribution to (46) coming from the two-particle reduced density matrix is zero. This is due to the fact that in GP theory Theorem 1 implies $\frac{d}{dt}\langle \phi | a | \phi \rangle = 0$. Therefore, all terms involving $\frac{d}{dt}\langle \phi | a | \phi \rangle$ cancel also in the calculation of higher moments, see Appendix B.

# 6 Mean-Field violation of angular momentum conservation

In this section we set $A = L_z$, where

$$
L_z = \sum_{j=1}^{N} l_{z_j},
\tag{47}
$$

is the total angular momentum operator about the $z$-axis with $l_z = x p_y - y p_x$. Now consider a cylindrically symmetric external potential $V(\mathbf{r})$, such that $[H, L_z] = 0$. Using the orbital

$$
\phi(\mathbf{r}, 0) = \left( \frac{1}{\pi \sigma_0^2} \right)^{D/4} e^{-\frac{(\mathbf{r} - \mathbf{r}_0)^2}{2\sigma_0^2}} e^{i p_0 x},
\tag{48}
$$

we evaluate Eqs. (45-46). For $\mathbf{r}_0 = (x_0, y_0)$ in $D = 2$ and $\mathbf{r}_0 = (x_0, y_0, z_0)$ in $D = 3$ dimensions the result is

$$
\frac{d}{dt}\langle L_z \rangle_{GP} = 0
\tag{49}
$$

$$
\frac{d}{dt}\langle L_z^2 \rangle_{GP} = -N\lambda \frac{p_0 x_0}{(2\pi\sigma_0^2)^{D/2}} \neq 0
\tag{50}
$$

Equation (49) expresses the fact that the expectation values of conserved one-body operators are time-independent in the GP mean-field approximation, see Eq. (41) and Theorem 1. However, Eq. (50) is non-zero and constitutes an explicit violation of angular momentum conservation, $[H, L_z] = 0$. Higher moments, e.g. $\langle L_z^3 \rangle_{GP}$ can also be evaluated and are also time-dependent, see Appendix B.

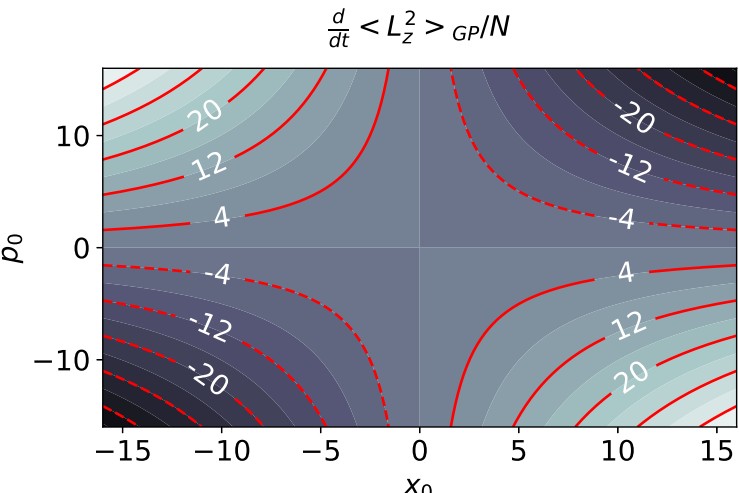

Figure 2: **Surface of angular momentum violations.** Shown is a contour plot of the time-derivative of the second moment of $L_z/N$ in Gross-Pitaevskii theory, $\frac{d}{dt}\langle L_z^2 \rangle_{GP}/N$, see Eq. (51), as a function of the initial displacement $x_0$ and the initial momentum $p_0$ (all other parameters constant $\lambda = 1$, $\sigma_0 = 1$, $D = 2$). The correct value for an angular momentum conserving theory is zero. $\frac{d}{dt}\langle L_z^2 \rangle_{GP}/N$ can take on any value between $-\infty$ and $+\infty$. Thus, the violation of angular momentum conservation can be arbitrarily large at any (nonzero) interaction strength. All units are dimensionless.

## 6.1 Parametric dependence

First we point out that the violation of angular momentum conservation in Eq. (50) originates from $\frac{d}{dt}\langle l_z^2 \rangle$, i.e. from the one-body part of (13), not the two-body part. The contribution to $\frac{d}{dt}\langle L^2 \rangle$ from the two-body part of (13) is precisely zero here.

Also note that the expression (50) scales linearly with the GP interaction parameter $\lambda$. Thus, for $\lambda = 0$, when the single-particle Schrödinger equation is recovered, the violation vanishes as expected. Remarkably, the violation (50) is not a finite $N$ effect. A popular limit to analyze the properties of infinite trapped systems is to keep $\lambda$ constant and to let $N \to \infty$. However, in this limit (50) diverges, because even the violation per particle

$$\frac{d}{dt}\langle L_z^2 \rangle_{GP}/N = -\lambda \frac{p_0 x_0}{(2\pi\sigma_0^2)^{D/2}} \qquad (51)$$

is constant, independent of $N$. For any given value of the interaction strength $\lambda$ the derivative of the violation $\frac{d}{dt}\langle L_z^2 \rangle_{GP}/N$ can be arbitrarily large, since Eq. (51) then defines a hyperbola between the parameters $x_0$ and $p_0$. This error surface is shown in Fig. 2 for $\lambda = \sigma_0 = 1$ in $D = 2$ dimensions. In other words, in GP theory the violation of angular momentum conservation can be arbitrarily large for any given value of the interaction strength $\lambda \neq 0$. From the expression (50) it can be seen that $\frac{d}{dt}\langle L_z^2 \rangle_{GP}$ scales linearly with the displacement of the wave packet $x_0$ from the center of the trap and the momentum $p_0$ along that direction. Displacements $y_0$ along the direction orthogonal to $p_0$ do not affect $\frac{d}{dt}\langle L_z^2 \rangle_{GP}$. However, they enter higher moments (which are also time dependent), see Appendix B.

The reason for this failure of GP theory is explained by Theorem (1). The space of allowed

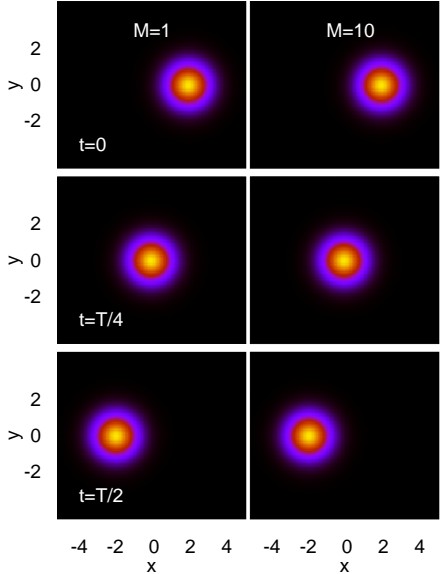

Figure 3: **Sloshing of a BEC in a harmonic trap.** Shown is the single-particle density $\rho(\mathbf{r}, t)$ using different numbers of orbitals. The BEC oscillates in a harmonic trap with $\omega_x = \omega_y = 1$. The initial state is the ground state of the trap displaced to $(x_0, y_0) = (2, 0)$. The interaction is weak with the ratio of interaction to kinetic energy of the initial state being 6%. Left: Gross-Pitaevskii ($M = 1$). Right: Many-body result using MCTDHB with $M = 10$ orbitals. There is no visible difference between the $M = 1$ and the $M = 10$ result. Interaction parameter $\lambda = 0.9$. Number of particles $N = 10$. Oscillation period $T = 2\pi$. All units are dimensionless.

variations $\{\delta\Psi\}$ of GP theory, see (41), is too small to accomodate the vector $L_z^2|\Psi\rangle$. Angular momentum can therefore only be conserved if the space of allowed variations $\{\delta\Psi\}$ is enlarged. Within the MCTDHB framework this entails using more orbitals in the ansatz (26). It is not possible to tell *a priori* how many more orbitals are needed to conserve angular momentum to an acceptable level, because this depends on the shape of the tangent space $\{\delta\Psi\}$ at $\Psi(t)$.

# 7   Sloshing of a BEC in a harmonic trap

In this section we provide a simple numerical example in $D = 2$ dimensions that illustrates the violation of angular momentum conservation by GP mean-field theory and the gradual restoration of angular momentum conservation on the many-body level.

## 7.1   Depletion and interaction energy

In order to quantify the degree to which angular momentum conservation is violated, we chose an extremely generic system: a BEC that oscillates in a harmonic trap in two spatial dimensions. According to (50) the violation $\frac{d}{dt}\langle L_z^2\rangle_{GP}$ is proportional to $N\lambda$. We therefore

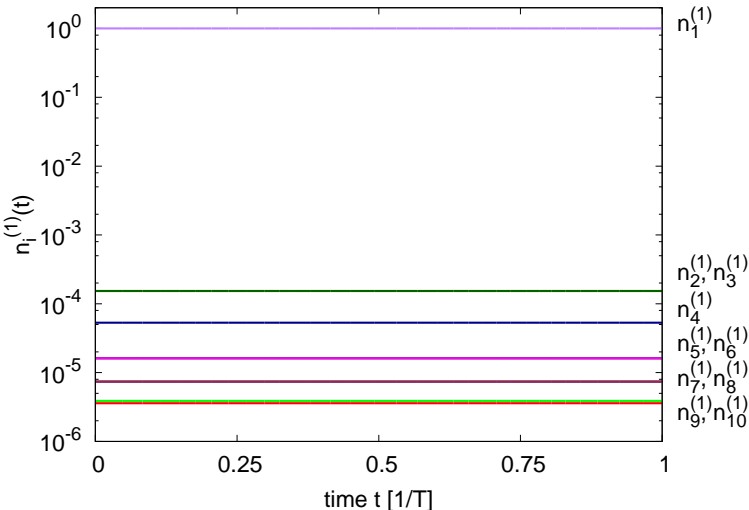

Figure 4: **Evolution of natural occupation numbers.** Shown are the natural orbital occupation numbers of the many-body simulation using $M = 10$ orbitals. The pairs of occupation numbers $n_2^{(1)}, n_3^{(1)}$ as well as $n_5^{(1)}, n_6^{(1)}$ and $n_7^{(1)}, n_8^{(1)}$ are quasi degenerate and the respective lines lie on top of each other. The condensate is only weakly depleted with the largest natural orbital occupations being $n_1^{(1)} = 99.95\%$ and $n_2^{(1)} = n_3^{(1)} = 0.015\%$. For the time shown here there is no visible change in the natural occupation numbers. The system remains condensed throughout. All units are dimensionless.

choose a fixed value $\lambda = 0.9$. As a trapping potential we use $V(\mathbf{r}) = \frac{1}{2}\omega^2(x^2 + y^2)$ with $\omega = 1$ and solve the many-body Schrödinger equation using the MCTDHB method for different numbers of orbitals $M$ in order to converge to the exact many-body result. To observe this convergence we use $N = 10$ bosons for the many-body simulations for which we use up to $M \leq 10$ orbitals. For the largest simulations, i.e. for $M = 10$ the Hilbert space then uses 92378 time-adaptive permanents as opposed to just one for $M = 1$.

The GP ground state energy per particle is $E_{GP}/N = 1.0675$ and the ratio of interaction energy to total energy is only about 6%, so the interaction energy is only small compared to the kinetic energy. The energy hardly changes when $M > 1$ orbitals are used: for $M = 2$ the ground state energy per particle is $E_{M=2}/N = 1.0673$ and for $M = 9, 10$ one finds $E_{M=10}/N = 1.0666$. The natural occupations converge to values near the GP result. For $M = 9, 10$ one finds only a small depletion of the first natural orbital with $n_1^{(1)}/N = 0.99959$. One would therefore expect the GP mean-field to be a good approximation to the true many-body results.

## 7.2 Evolution of the single-particle density and the natural occupations

For each value of $M$ we displace the respective ground state from the origin to $(x_0, y_0) = (2, 0)$ and propagate the many-body Schrödinger equation using the MCTDHB method. The condensate sloshes back and forth in the trap with a time period $T = 2\pi$, as shown in Fig. 3. The two columns show the single particle density $\rho(\mathbf{r}, t)$ computed using $M = 1$ and $M = 10$ orbitals at times $t = 0, t = T/4$ and $t = T/2$. There is no visible difference between the GP

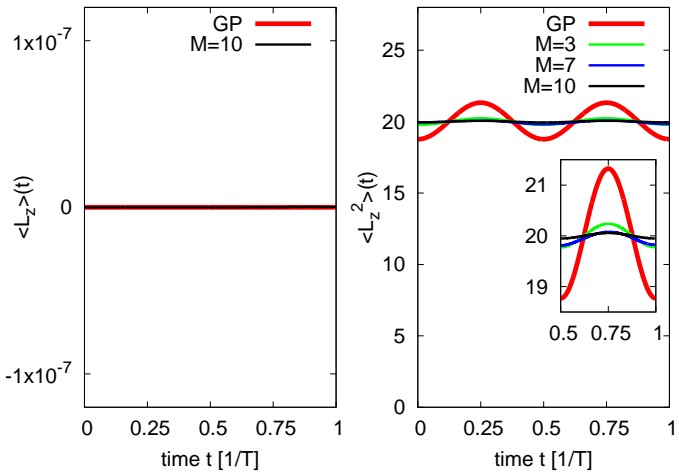

Figure 5: **Conservation of angular momentum.** Shown are the expectation values $\langle L_z \rangle(t)$ (left) and $\langle L_z^2 \rangle(t)$ (right) for the dynamics shown in Fig. 1. The Hamiltonian commutes with $L_z$ and hence in the exact many-body Schrödinger dynamics $\langle L_z \rangle(t)$ and $\langle L_z^2 \rangle(t)$ are time-independent. Gross-Pitaevskii mean-field theory has a stationary mean-value $\langle L_z \rangle(t)$, but violates angular momentum conservation through a time-dependent $\langle L_z^2 \rangle(t)$. The Gross-Pitaevskii result oscillates with an amplitude of about 13% of the absolute value instead of being constant. Many-body MCTDHB simulations using $M = 3, 7, 10$ approach the exact constant result for increasing numbers of orbitals. The oscillations for the $M = 10$ MCTDHB result are only about 0.5 % of the absolute value, i.e. twenty-six times better. All units are dimensionless.

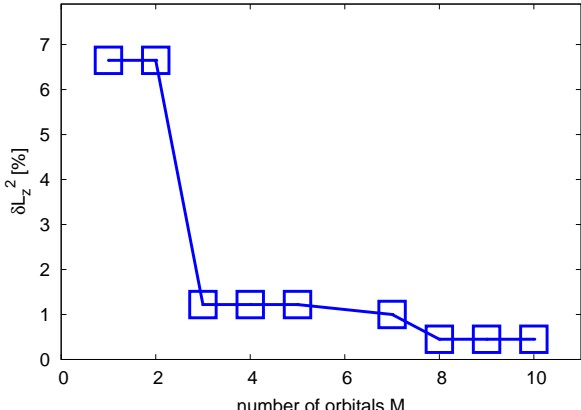

Figure 6: **Convergence towards the exact result.** Shown is $\delta L_z^2$, the largest relative deviation of $\langle L_z^2 \rangle (t)$ from the time averaged value, $\overline{\langle L_z^2 \rangle}$ as a function of the number of orbitals used in the computation. For increasing numbers of orbitals and thus larger many-body Hilbert spaces the oscillation amplitude decays. See text for details. All units are dimensionless.

mean-field and the many-body result. The single-particle density is essentially the same for GP mean-field theory and the $M = 10$ many-body result. There is practically no change in the natural occupation numbers during the dynamics. This is shown in Fig. 4: not even the smallest natural occupations vary significantly during the first oscillation of the condensate in the trap. To a very good approximation the condensate remains condensed.

## 7.3   Conservation of angular momentum

We now turn to the conservation of angular momentum. The trap is cylindrically symmetric, therefore the exact solution of the Schrödinger equation conserves angular momentum. In the left panel of Fig. 5 it can be seen that the expectation value $\langle L_z \rangle (t)$ is precisely zero and conserved for both by GP mean-field theory as well as the many-body result ($M = 10$), just as is expected based on the discussion in sections 4.3 and 4.4.

The right panel of Fig. 5 shows the time evolution of the expectation value $\langle L_z^2 \rangle (t)$ for different numbers of orbitals. The GP result for $\langle L_z^2 \rangle (t)$ oscillates at twice the trap frequency within a range of about 13% of its average value. Maxima of $\langle L_z^2 \rangle (t)$ correspond to the times when the condensate has maximal momentum, i.e. at the origin of the trap, whereas the minima correspond to the turning points. The GP dynamics thus still violates the conservation of angular momentum. In fact, the violation is quite strong (13%), despite the fact that the interaction energy accounts for merely 6% of the total energy and the depletion is just $\approx 4 \times 10^{-4}$.

For the MCTDHB simulations with $M > 1$ it can be seen that for increasing $M$ the amplitude of the oscillations decreases and approaches a constant, i.e. angular momentum conservation is violated less and less, the larger $M$ is. The reason is that the space of allowed variations gets larger and larger with increasing $M$ and therefore a larger fraction of the vector $|L_z^2 \Psi \rangle$ lies in the space of allowed variations. According to Theorem (1) the magnitude of the violation of angular momentum conservation then decreases. The violation can be quantified

by defining the time average $\overline{\langle L_z^2 \rangle} = \frac{1}{T} \int_0^T \langle L_z^2 \rangle(t) dt$ and the largest relative deviation

$$\delta L_z^2 = \max_{t \in [0,T]} \frac{|\langle L_z^2 \rangle - \overline{\langle L_z^2 \rangle}|}{\overline{\langle L_z^2 \rangle}}, \tag{52}$$

shown in Fig 6 as a function of the number of orbitals used in the computation. For increasing numbers of orbitals $\langle L_z^2 \rangle(t)$ approaches a constant, as required by angular momentum conservation. In other words the results converge with increasing $M$ to the exact result. However, even when $M = 10$ orbitals are used, there still remains a visible oscillation with an amplitude of about 0.5% of $\overline{\langle L_z^2 \rangle}$. As the permanents used in the numerical computations are determined from the variational principle the many-body results cannot be improved by choosing a more suited basis set of the same size. This is equally true for GP theory as well as in the many-body simulations. Hence angular momentum conservation is only possible on the many-body level. The significant size of the violation observed, implies that $\langle L_z^2 \rangle$ is a sensitive probe for beyond mean-field effects if the full many-body Hamiltionian satisfies $[H, L_z] = 0$.

## 8  Mean-Field violation of momentum conservation

In this section we provide an example for a dynamic violation of total momentum conservation – or equivalently translational symmetry – by the GP mean-field. We set $A = P$, where

$$P = \sum_{j=1}^N p_j, \tag{53}$$

is the total momentum operator with $p$ the single particle momentum operator. We work in $D = 1$ dimensions and consider free space, i.e. $V(x) = 0$ in (2), such that $[H, P] = 0$. We introduce a momentum chirped orbital

$$\phi(x, 0) = \left( \frac{1}{\pi \sigma_0^2} \right)^{1/4} e^{-\frac{(x - x_0)^2}{2\sigma_0^2}} e^{ip_0 x^2} \tag{54}$$

and evaluate Eqs. (45-46). The result of the calculation is

$$\frac{d}{dt} \langle P \rangle_{GP} = 0 \tag{55}$$

$$\frac{d}{dt} \langle P^2 \rangle_{GP} = N\lambda \frac{2p_0}{\sqrt{2\pi\sigma_0^2}} \neq 0. \tag{56}$$

As for angular momentum, we see that the mean-value of total momentum $\langle P \rangle_{GP}$ is stationary as can be understood on the basis of Eq. (41) and Theorem 1. However, Eq. (56) is non-zero and constitutes an explicit violation of momentum conservation, by the GP mean-field. Similar to the displacement in real space in Eqs. (49-50) the violation of momentum conservation in (56) is linear in $p_0$. Also higher moments can be evaluated. These are also time-dependent and thus contribute to the violation of momentum conservation in GP theory, please see Appendix B. As a numerical example, we consider a BEC which is released from a harmonic trap in 1D. A detailed many-body analysis of this problem with a closely related focus can be be found in [38]. We only provide the mean-field result to show the extent to which the GP dynamics violates the conservation of momentum.

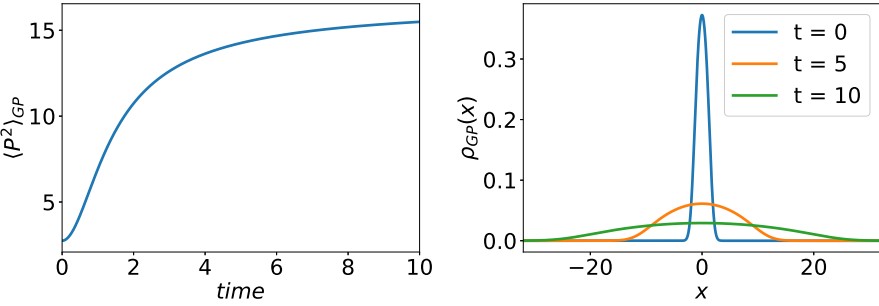

Figure 7: **Mean-field violation of momentum conservation.** A condensate is released from a harmonic trap in one spatial dimension into free space. Left: shown is the Gross-Pitaevskii mean-field result ($\lambda = 5$) for $\langle P^2 \rangle$. Momentum conservation requires $\langle P^2 \rangle$ to be constant in time. The mean-field dynamics violates momentum conservation. Right: corresponding one particle density at different times. See text for details. All quantities shown are dimensionless.

The BEC is prepared in the GP ground state of the potential $V(x) = \frac{1}{2}x^2$ at an interaction strength $\lambda = 5$ on an interval $[-64, 64]$. At time $t = 0$ the trap switched off and the GP equation is propagated in time. The density expands as the BEC is released from the trap. Fig. 7 (left) shows the time evolution of $\langle P^2 \rangle$. It can clearly be seen that $\langle P^2 \rangle$ is not constant in time, i.e. the GP dynamics strongly violates the conservation of total momentum $[H, P] = 0$. The right panel of Fig. 7 shows the corresponding single-particle density of the GP solution at different times.

We would like to stress that the kinetic energy of the system is not conserved in the exact dynamics. In a many-body system the kinetic energy is not proportional to $P^2$. So while the exact result for $\langle P^2 \rangle$ is to be constant in time, $\langle \sum_{i=1}^{N} p_i^2 \rangle$ is time-dependent. We have included a calculation that illustrates this point in detail in Appendix C.

## 9  Conclusion

In this work we have provided a rigorous explanation why conservation laws in BEC dynamics can only be satisfied on the many-body level. Specifically we have shown that the popular GP mean-field violates angular momentum and momentum conservation in analytical and numerical examples. For an observable $A$ that is conserved $[H, A] = 0$ in the exact dynamics, the GP ansatz wave function is flexible enough to keep the first moment $\langle A \rangle$ constant in time, but not the higher moments $\langle A^n \rangle$ for $n > 1$. We have shown that this is equivalent to a violation of the conservation law $[H, A] = 0$. This is experimentally verifiable by measuring $A$: the distribution of measured values $P(\mathcal{A})$ depends on all moments $\langle A^n \rangle$ of $A$ and a single time-dependent moment makes $P(\mathcal{A})$ time-dependent. Using the MCTDHB method and successively more general ansatz wave functions, we have shown how this improves the accuracy to which conservation laws are satisfied, as required by the exact dynamics. In summary, satisfying conservation laws in the time dynamics requires many-body theory even for weakly interacting systems at low depletion. It will be particularly interesting to investigate

the implications of these result for the creation of vortices in Bose-Einstein condensates.

## Acknowledgements

We thank Ofir Alon, Alexej Streltsov, Raphael Beinke and Axel Lode for constructive discussions. We thank the HLRS – High Performance Computing Center Stuttgart as well as the Vienna Scientific Cluster for providing computational resources for the many-body simulations.

**Funding information**   Financial support was provided by the ERC advanced grant 'QuantumRelax', Project ID: 320975.

## A    Time-indepedence of moments of conserved observables

We provide an alternative proof for the fact that conserved observables have constant moments. The proof is adopted from the textbook [39] (chapter 3 D.2.c. property (iii)).

**Theorem 2.** *Let $\Psi$ be a wave function, $H$ a Hamiltonian, $\frac{\partial H}{\partial t} = 0$, and $A$ an observable, $\frac{\partial A}{\partial t} = 0$ and $[H, A] = 0$, then $\frac{d}{dt}\langle\Psi|A^n|\Psi\rangle = 0$ for all $n$.*

*Proof.* Since $[H, A] = 0$, $H$ and $A$ have a common set of eigenfunctions $\{|E_j, \lambda_k\rangle\}$, i.e.

$$H|E_j, \lambda_k\rangle = E_j|E_j, \lambda_k\rangle \tag{57}$$
$$A|E_j, \lambda_k\rangle = \lambda_k|E_j, \lambda_k\rangle. \tag{58}$$

Expanding $|\Psi(t)\rangle$ in this basis yields $|\Psi(t)\rangle = \sum_{j,k} C_{j,k}(t)|E_j, \lambda_k\rangle$ with $C_{j,k}(t) = C_{j,k}(0)\exp(-iE_j t)$ and therefore

$$\langle A^n\rangle(t) = \sum_{j,k} |C_{j,k}(t)|^2 \lambda_k^n = \sum_{j,k} |C_{j,k}(0)|^2 \lambda_k^n = \langle A^n\rangle(0), \tag{59}$$

i.e. $\frac{d}{dt}\langle\Psi|A^n|\Psi\rangle = 0$ for all $n$.    □

## B    Mean-Field violation of conservation laws by $\langle A^3\rangle_{GP}$

In the main text we showed that $[H, A] = 0$ is dynamically violated in GP theory for an operator A defined in 6 by evaluating $\frac{d}{dt}\langle A^2\rangle$. Here we show that also higher moments violate $[H, A] = 0$. The operator $A^3$ can be written as

$$A^3 = \sum_i^N a_i^3 + 3\sum_{i\neq j}^N a_i^2 a_j + \sum_{i\neq j\neq k}^N a_i a_j a_k. \tag{60}$$

                                                                    

In second quantization

$$
\begin{aligned}
A^3 &= \sum_{ij} b_i^\dagger b_j \langle \phi_i | a^3 | \phi_j \rangle \\
&+ 3 \sum_{ijkl} b_i^\dagger b_j^\dagger b_k b_l \langle \phi_i | a^2 | \phi_k \rangle \langle \phi_j | a | \phi_l \rangle \\
&+ \sum_{ijklmn} b_i^\dagger b_j^\dagger b_k^\dagger b_l b_m b_n \langle \phi_i | a | \phi_l \rangle \langle \phi_j | a | \phi_m \rangle \langle \phi_k | a | \phi_n \rangle
\end{aligned} \tag{61}
$$

and therefore

$$
\begin{aligned}
\langle A^3 \rangle &= \sum_{ij} \rho_{ij} \langle \phi_i | a^3 | \phi_j \rangle \\
&+ 3 \sum_{ijkl} \rho_{ijkl} \langle \phi_i | a^2 | \phi_k \rangle \langle \phi_j | a | \phi_l \rangle \\
&+ \sum_{ijklmn} \rho_{ijklmn} \langle \phi_i | a | \phi_l \rangle \langle \phi_j | a | \phi_m \rangle \langle \phi_k | a | \phi_n \rangle
\end{aligned} \tag{62}
$$

with the three-particle reduced density matrix elements

$$
\rho_{ijklmn} = \langle \Psi | b_i^\dagger b_j^\dagger b_k^\dagger b_l b_m b_n | \Psi \rangle. \tag{63}
$$

Using the ansatz (38) it follows from Theorem 1 that for GP theory $\frac{d}{dt} \langle \phi | a | \phi \rangle = 0$ and therefore the 3-body contribution drops out and one arrives at

$$
\frac{d}{dt} \langle A^3 \rangle_{GP} = N \frac{d}{dt} \langle \phi | a^3 | \phi \rangle + 3N(N-1) \langle \phi | a | \phi \rangle \frac{d}{dt} \langle \phi | a^2 | \phi \rangle \tag{64}
$$

For a specific example how higher moments also violate angular momentum conservation we work in $D = 2$ dimensions and use the initial state

$$
\phi(x, y, 0) = \left( \frac{1}{\pi \sigma_0^2} \right)^{1/2} e^{-\frac{(x-x_0)^2 + (y-y_0)^2}{2\sigma_0^2}} e^{ikx}. \tag{65}
$$

Then in order to calculate (64) for $A = L_z$ we evaluate

$$
\langle \phi | l_z | \phi \rangle = -k y_0 \tag{66}
$$

$$
\frac{d}{dt} \langle \phi | l_z | \phi \rangle = 0 \tag{67}
$$

$$
\frac{d}{dt} \langle \phi | l_z^2 | \phi \rangle = -\lambda \frac{k x_0}{2\pi \sigma_0^2} \tag{68}
$$

$$
\frac{d}{dt} \langle \phi | l_z^3 | \phi \rangle = \lambda \frac{3k^2 x_0 y_0}{2\pi \sigma_0^2} \tag{69}
$$

and therefore

$$
\frac{d}{dt} \langle L_z^3 \rangle_{GP} = N^2 \lambda \frac{3k^2 x_0 y_0}{2\pi \sigma_0^2} \tag{70}
$$

Thus, $\langle L_z^3 \rangle$ is time-dependent in GP theory and contributes just like $\langle L_z^2 \rangle$ to the violation of angular momentum conservation.

Similarly we evaluate the time derivative of the third moment of the total momentum operator in GP theory for the chirped orbital (54). In order to calculate (64) for $A = P$ we evaluate

$$\langle \phi | p | \phi \rangle = 2 p_0 x_0 \tag{71}$$

$$\frac{d}{dt} \langle \phi | p | \phi \rangle = 0 \tag{72}$$

$$\frac{d}{dt} \langle \phi | p^2 | \phi \rangle = \frac{2 \lambda k p_0}{\sqrt{2 \pi \sigma_0^2}} \tag{73}$$

$$\frac{d}{dt} \langle \phi | p^3 | \phi \rangle = \frac{12 \lambda p_0^2 x_0}{\sqrt{2 \pi \sigma_0^2}} \tag{74}$$

Thereby one arrives at

$$\frac{d}{dt} \langle P^3 \rangle_{GP} = N^2 \lambda \frac{12 p_0^2 x_0}{\sqrt{2 \pi \sigma_0^2}} \tag{75}$$

As before, $\langle P^3 \rangle_{GP}$ is time-dependent and contributes just like $\langle P^2 \rangle$ to the violation of momentum conservation. Note how the contributions from the two- and three-particle reduced density matrices add up to precisely $N^2$ times the results (69) and (74). The evaluation of expressions for higher moments becomes more and more cumbersome. However, there is no conceptual novelty in the evaluation of ever higher moments. Therefore, we stop here at the third moment.

## C  Kinetic energy non-conservation

We show in an example that for interacting translationally invariant systems momentum is conserved, but the kinetic energy is not. We show this in order to prevent a possible confusion arising from the time-independece of $\langle P^2 \rangle$ that follows from the conservation of $P$.

Consider two particles of the same mass and a purely distance dependent two-body interaction potential in $D = 1$ dimensions, $W(r)$ with $r = |x_1 - x_2|$. The particles are described by a Hamiltonian $H = T + W$, where

$$T = \frac{p_1^2}{2} + \frac{p_2^2}{2} \tag{76}$$

is the kinetic energy. For the following we note that

$$\text{sign}(x_1 - x_2) = \frac{x_1 - x_2}{|x_1 - x_2|}. \tag{77}$$

To see that $P$ defined in Eq. (53) is conserved we note that $[P, T] = 0$, since $[p_i, p_j] = 0$. Furthermore $[P, W] = 0$ as can be seen by adding the commutators

$$[p_1, W] = -i \frac{dW(r)}{dr} \text{sign}(x_1 - x_2) \tag{78}$$

$$[p_2, W] = +i \frac{dW(r)}{dr} \text{sign}(x_1 - x_2). \tag{79}$$

Therefore $[P, H] = 0$ and $P$ is conserved. It follows immediately from Eq. (9) that $\langle P^2 \rangle$ is time-independent.

To see that the kinetic energy is not conserved we evaluate

$$\left[p_1^2, W\right] = -\frac{d^2 W(r)}{dr^2} - 2i\frac{dW(r)}{dr}\,\text{sign}(x_1 - x_2)p_1 \tag{80}$$

$$\left[p_2^2, W\right] = +\frac{d^2 W(r)}{dr^2} + 2i\frac{dW(r)}{dr}\,\text{sign}(x_1 - x_2)p_2 \tag{81}$$

and therefore, using $[T, H] = [T, W]$, we find that the kinetic energy is not conserved

$$[T, H] = -i\frac{dW(r)}{dr}\,\text{sign}(x_1 - x_2)(p_1 - p_2) \neq 0. \tag{82}$$

This expresses the fact that in a many-body system the operator for the square of the total momentum is not proportional to the kinetic energy:

$$P^2 = \left(\sum_{i=1}^{N} p_i\right)^2 = \sum_{i=1}^{N} p_i^2 + 2\sum_{i<j} p_i p_j = T + 2\sum_{i<j} p_i p_j. \tag{83}$$

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
