# Peer review of "Conserving symmetries in Bose-Einstein condensate dynamics requires many-body theory"

_SciPost Physics_

## Round 1 · Referee Report · Anonymous (Referee 1) · 2018-4-25

Strengths

1 Clear and pedagogical
2 Quantify in an interesting case the deviations from the exact result for the second momentum of L_z

Weaknesses

1 Emphasis not clear, or misleading according me.
2 No parallel discussion of the behaviour of the expectation values of higher momenta of momentum p is provided.

Report

The Authors study the conservation of angular momentum for weakly interacting Bose gases. After setting the formalism, they write a relation for the time evolution of the expectation values of L_z^n, showing that for n=2 one has that such time derivative is not zero. They then show that adding orbitals these deviations decrease.

My reaction to the paper is two-fold: from one side I think the paper is remarkably clear, even pedagogical, with a true effort to be self-contained. From the other side I do not agree with the emphasis of the paper. Let me explain better this point: when one does mean-field and a quantity is conserved (like the energy), one knows it may be different from the exact one. Nevertheless it may be a good approximation – however there are quantities that cannot be captured in mean-field, like correlations. So, there are quantities which are simply not exactly or badly captured by mean-field. The Authors shows that (d/dt) <L_z^n> is zero for n=1, and not zero for n=2. I conclude that the angular momentum is conserved, but that higher momenta of it are not. Actually I am even surprised that the error for n=2 is relatively small like the one shown in Fig.4. Let me come to the main point: suppose that one computes (d/dt) <p_z^n> and shows that it is zero for n=1, and not zero for n=2. One would entitle the paper “Conservation of momentum in Bose-Einstein condensates requires many-body theory”? Actually, my question is: do it is true that (d/dt) <p_z^n> is zero for n=1, and zero also for n larger that 2, differently from the case considered in the paper? Do p and L_z are different for the purposes of the paper? After all, translational invariance would require exact conservation of all momenta of p (when there is no external potential), and I do not see how the argument would be different. In other words the Authors could/should make similar computations and considerations for the momentum p and the reader would like to understand why the two cases may be different (if they are). I think that such a discussion would be useful for the clarity and the substance of the paper.

As a Referee, I think is completely up to the Authors to choose the title and the emphasis of their paper, but I am also concerned about the fact that the title would lead the reader to think that angular momentum is not conserved in mean-field, while it is. Are the (expectation values of the) higher momenta of L_z that are not conserved. So, when the Authors say “However, equations (37) and (39) are generally not zero and thus constitute explicit violations of the conservation of angular momentum in two- and three-dimensional GP theory”, I would say that “constitute explicit violations of the conservation of higher momenta (n \geq 2) of angular momentum”. If no discussion of the corresponding results for momentum p and no change of emphasis is done (including the title and the abstract), I do not think the paper should be published. For this reason I suggest to the Authors major revisions according the lines discussed in this report.

Requested changes

1 Provide a discussion of the corresponding results for the momentum p in the translational invariant case
2 Reconsider the emphasis of the presentation

  • validity: high
  • significance: ok
  • originality: good
  • clarity: top
  • formatting: excellent
  • grammar: good

Author:  Kaspar Sakmann  on 2018-12-13  [id 383]

(in reply to Report 1 on 2018-04-25)
Category:
remark

added pdf of response to referee 1.

Attachment:

response_ref1.pdf

Author:  Kaspar Sakmann  on 2018-12-13  [id 382]

(in reply to Report 1 on 2018-04-25)
Category:
reply to objection

Referee 1 writes: ''1 - Emphasis not clear, or misleading according me. ... The Authors shows that $\frac{d}{dt}\langle L_z^n\rangle$ is zero for $n=1$, and not zero for $n=2$. I conclude that the angular momentum is conserved, but that higher momenta of it are not.''

Answer: There appears to be a misunderstanding: for the time-indepdendent case mean-field violations of conservation laws have been known for a long time. We show the same is true for the time-dependent case, explain the origin rigorously and provide quantitative predicitions. Referee 1's conclusion that angular momentum is conserved if $\frac{d}{dt}\langle L_z\rangle=0$, but $\frac{d}{dt}\langle L_z^n\rangle\neq0$ for $n>1$, is not correct in quantum mechanics. Let us explain.

  1. Observables in quantum mechanics are represented by hermitian operators. Angular momentum is the operator $L_z$, it is not the expectation value $\langle L_z\rangle$.

  2. The condition for an observable $A$ to be conserved is $[H,A]=0$.

  3. $\frac{d}{dt}\langle L_z\rangle=0$ follows from $[H,L_z]=0$. It is a necessary, but not a sufficient condition for the conservation of angular momentum. See points 4, 5, 6.

  4. It is incorrect to conclude that an observable $A$ is conserved solely on the basis that $\frac{d}{dt}\langle A\rangle=0$, as done by referee 1. To see this consider a single particle at rest in free space. The Hamiltonian is $H=p^2/2$. The observable $x$ is not conserved: $[H,x]\neq0$. Nevertheless $\frac{d}{dt}\langle x\rangle=0$. Please see the new section 2.5 of the revised manuscript for more details.

  5. $[H,L_z]=0$ implies $\frac{d}{dt}\langle L_z^n\rangle=0$ for all $n$, as shown in Eq. (9). Another proof, taken from a textbook, can be found in Appendix A. The probability distribution of measurements of $L_z$ is only stationary if $\frac{d}{dt}\langle L_z^n\rangle=0$ for all $n$. This is proven in Eq. (18). Any $\frac{d}{dt}\langle L_z^n\rangle\neq 0$ is a violation of $[H,L_z]=0$ and is experimentally detectable by measuring $L_z$, see section the new section 2.5 and the new Fig. 1.

  6. Ground state mean-field solutions have been known to violate conservation laws since 1963 (Hartree-Fock) and at least since 1975 (GP mean-field). Even the methods to restore these broken symmetries have been textbook material since at least 1980. It is a well-established fact that the GP mean-field violates conservation laws. We only provide new results for the time-dependent case. See the introduction for the historical context.

  7. We provide an explanation why satisfying conservation laws in the dynamics is only possible on the many-body level, see section 4.3 and 4.4, as well as parametric dependencies of the violations and how to fix the problem. None of this has been published before.

The above points are discussed in detail in the new version of the manuscript, including proofs, examples and references. Satisfying conservation laws in BEC dynamics requires many-body theory. We have changed the abstract to emphasize that we explain why this is the case.

Referee 1 writes: ''Actually I am even surprised that the error for n=2 is relatively small like the one shown in Fig.4. [now Fig.6]}''

Answer: 1. $\langle L_z\rangle=0$ is exactly constant in the dynamics. Even in the numerics at a level of the numerical precision $<10^{-8}$. In contrast, $\langle L_z^2\rangle$ varies over $\pm 13\%$, a difference of at least seven(!) orders of magnitude.

  1. As stated in the abstract, we chose very weak interaction strength and practically no depletion (<$5\times 10^{-4}$) to make the conditions as favorable as possible for the GP mean-field. Nevertheless, angular momentum conservation is substantially violated.

  2. Arbitrarily strong violations: $\frac{d}{dt}\langle L_z^2\rangle$ can take on any value between $-\infty$ and $\infty$ depending on the values chosen for $x_0, \sigma_0, p_0$, see Eq. (50). The violation grows linearly with the initial displacement from the center of the trap $x_0$, linearly with the initial momentum $p_0$, linearly with the GP nonlinearity parameter $\lambda$ and so on. It is no problem at all to find larger violations. To illustrate this fact we have included a new Fig. 2 to show these parametric dependencies.

  3. Many experiments work in the opposite limit where the kinetic energy is much smaller than the interaction energy. The violation grows with the interaction strength, see Eq. (50).

Referee 1 writes: ''Let me come to the main point: ... Actually, my question is: do it is true that $(d/dt) <p_z^n>$ is zero for $n=1$, and zero also for n larger that 2, differently from the case considered in the paper? Do $p$ and $L_z$ are different for the purposes of the paper? After all, translational invariance would require exact conservation of all momenta of $p$ (when there is no external potential), and I do not see how the argument would be different. In other words the Authors could/should make similar computations and considerations for the momentum $p$ and the reader would like to understand why the two cases may be different (if they are). I think that such a discussion would be useful for the clarity and the substance of the paper.''

Answer: 1. We thank the referee for suggesting to look at the conservation of linear momentum as an additional example. Our calculations are general. There is no conceptual difference between angular and linear momentum conservation. We picked violations of angular momentum conservation out of personal preference. The GP dynamics also violates momentum conservation. We have included a specific example for a dynamic violation of momentum conservation by mean-field in a new section 8.

  1. Following referee 1's suggestion we now first discuss the (non-)conservation of general one-body operators $A$. Later we specialize to $A=L_z$ and $A=P$ and provide specific examples. However, we have kept the main focus of the paper on angular momentum.

Referee 1 writes: ''... I am also concerned about the fact that the title would lead the reader to think that angular momentum is not conserved in mean-field, while it is. Are the (expectation values of the) higher momenta of $L_z$ that are not conserved. So, when the Authors say "However, equations (37) and (39) are generally not zero and thus constitute explicit violations of the conservation of angular momentum in two- and three-dimensional GP theory'', I would say that "constitute explicit violations of the conservation of higher momenta ($n \geq 2$) of angular momentum''.''

Answer: As discussed above, Referee 1's conclusion that angular momentum is conserved when $\frac{d}{dt}\langle L_z\rangle=0$, but $\frac{d}{dt}\langle L_z^n\rangle=0$ for $n>1$ is not correct.

  1. The condition for angular momentum conservation is $[H,L_z]=0$, which implies $\frac{d}{dt}\langle L_z^n\rangle=0$ for all $n$. Therefore angular momentum is not conserved in the GP mean-field dynamics.

  2. The full probability distribution of the measured values of $L_z$ involves all moments $\langle L_z^n\rangle$. If any of these is time-dependent, so is this probability distribution. See Eq. (18) as well as the new sections 2.4 and 2.5.

  3. In quantum mechanics it is not correct to conclude that an observable is conserved only because its expectation value $\langle A\rangle$ is time-independent. It is important to realize that even for observables $A$ that are not conserved, $[H,A]\neq0$, one can have $\frac{d}{dt}\langle A\rangle=0$. We have now included a single particle counter example, for details see the new section 2.5.

Conclusion: Referee 1 concludes (incorrectly) from $\frac{d}{dt}\langle L_z\rangle=0$ that angular momentum is conserved by the GP mean-field dynamics, whereas for conservation of $L_z$ in quantum mechanics all moments $\langle L_z^n\rangle$ have to be time-independent, which is not the case. Stationary mean-field violations of conservation laws have been long known. We treat the time-dependent case. We explain from first principles why conservation laws can only be satisfied on the many-body level in the dynamics. In order to avoid further confusion we have included the relevant background material that illustrates these points. Furthermore, we included an example for the violation of linear momentum conservation by the GP mean-field, as was asked for by Referee 1 as well as expressions for higher order momenta.

---

## Round 1 · Referee Report · Anonymous (Referee 2) · 2018-4-26

Strengths

1- the idea of comparing explicitly the GP approach with its multi-orbital extension for a relevant quantity as the angular momentum L_z and its fluctuations L_z^2

Weaknesses

1 - the aim of the manuscript - as highlighted very strongly also in the title and abstract - is misleading to say the least.
Indeed the authors seem to infer that it is surprising that
the dynamics of the fluctuations of the angular momentum is not taken into account properly by GP approach to a BEC.

However the fact that GP or better mean field approaches cannot, in general, properly describe fluctuations (their static as well) is well known.

2 - (given 1-) only the dipole mode with L_z=0 is considered. And only the second order L_z^2. Further cases are needed.

3 - (given 1-) Comments on the use of Bogolyubov theory or linearised GP approach above the ground state in the linear regime to determine fluctuations - as done in literature - are completely absent.

Report

From the title and the abstract the reader infer that the aim of the present manuscript is to show that the mean field Gross-Pitaevskii (GP) description of a Bose-Einstein condensate (BEC) is unable to properly take into account angular momentum conservation.

Reading the manuscript however it turns out that the author do not mean the conservation of the angular momentum <L_z> — which indeed the author show to be well account for within GP — but of the absence of conservation of the higher order momenta of the the angular momentum distribution. In particular they focus on the dynamics of <L_z^2>, i.e., the dynamics of the fluctuation of the angular momentum. They show how only going beyond GP — they use for their purpose a multi-orbital approach — it is possible to obtain the proper dynamics, which in the case study correspond to no-evolution, i.e., conservation.

However the fact the GP approach is unable in general to properly describe fluctuations is well known. It has nothing to do with BEC, but it is true for any mean-field approach. It would be rather surprising the opposite.

Therefore (in the context of cold gases) no one would even use GP to describe not only the dynamics, but also the static value (as also the author find although without mentioning it, see Fig.3 right panel) of fluctuations.

For the above reason, in its present form, the manuscript does not deserve publication -- and I would even dare to say that it should no be published -- anywhere.

However I think the results on the comparison between GP (or single orbital) and a multi-orbital approach in describing static and dynamics of fluctuations are rather interested once put in the proper context. Here below just a few points: 1. the fact that the error is not very large even in the dynamics is per se an interesting result (although pointing in the opposite direction of the author`s aim). 2. the fact that the time average of the fluctuations calculated within GP reproduces the “exact” value of the fluctuations is interested. It is related to the use of the dynamics of fluctuations above GP equations, for e.g. determine number fluctuations, angular momentum fluctuations, condensate depletion, … 3. The needed number of orbital to recover conservation of L_z^2 is also an interesting information, at least once a few other momenta (as L_z^3 and L_z^4) are analysed. How is the number of needed multi orbital growing? Does there exist any scaling or saturation? 4. How the multi orbital approach compare, especially in the linear regime, with other more standard approaches to determine fluctuations, as, e.g., Bogolyubov approach?

Obviously my suggestion require a rather radical revision of the manuscript (including title and abstract) and a completely new point of view. However I believe that it is worth the effort.

Requested changes

1 - completely change the emphasis of the presentation of the results
2 - higher order momenta analysis (scaling, saturation of the orbital...)
3 - comment on or at least mention other works (mainly in linear response) in which fluctuations have been calculated.

  • validity: high
  • significance: low
  • originality: good
  • clarity: high
  • formatting: excellent
  • grammar: excellent

Author:  Kaspar Sakmann  on 2018-12-13  [id 385]

(in reply to Report 2 on 2018-04-26)
Category:
remark

added pdf of response to referee 2

Attachment:

response_ref2.pdf

Author:  Kaspar Sakmann  on 2018-12-12  [id 374]

(in reply to Report 2 on 2018-04-26)
Category:
reply to objection

Referee 2 writes: ''1 - the aim of the manuscript - as highlighted very strongly also in the title and abstract - is misleading to say the least \dots From the title and the abstract the reader infer that the aim of the present manuscript is to show that the mean field Gross-Pitaevskii (GP) description of a Bose-Einstein condensate (BEC) is unable to properly take into account angular momentum conservation. Reading the manuscript however it turns out that the author do not mean the conservation of the angular momentum $\langle L_z \rangle$ -- which indeed the author show to be well account for within GP -- but of the absence of conservation of the higher order momenta of the the angular momentum distribution.''

Answer: There appears to be a misunderstanding: for the time-independent case mean-field violations of conservation laws have been known for a very long time. We show the same is true for the time-dependent case, explain the origin rigorously and provide quantitative predicitions. Referee 2 states that angular momentum is conserved by the GP mean-field dynamics because the expectation value $\langle L_z \rangle$ is time-independent. This is not correct. In quantum mechanics $\frac{d}{dt}\langle L_z \rangle=0$ is only a necessary, but not a sufficient condition for the conservation of angular momentum. Let us explain.

  1. ''... the author do not mean the conservation of the angular momentum $\langle L_z \rangle$ ....'' Observables in quantum mechanics are represented by hermitian operators. Angular momentum is represented by the operator $L_z$, not by its expectation value $\langle L_z\rangle$.

  2. The condition for an observable $A$ to be conserved is $[H,A]=0$.

  3. $\frac{d}{dt}\langle L_z\rangle=0$ follows from $[H,L_z]=0$. It is a necessary, but not a sufficient condition for the conservation of angular momentum. See points 4, 5, 6.

  4. It is incorrect to conclude that an observable $A$ is conserved solely on the basis that $\frac{d}{dt}\langle A\rangle=0$. To see this consider a single particle at rest in free space. The Hamiltonian is $H=p^2/2$. The observable $x$ is {\it not} conserved: $[H,x]\neq0$. Nevertheless $\frac{d}{dt}\langle x\rangle=0$. Please see section 2.5 in the revised version for more details.

  5. $[H,L_z]=0$ implies $\frac{d}{dt}\langle L_z^n\rangle=0$ for all $n$, as shown in Eq. (9). Another proof, taken from a textbook, can be found in the Appendix of the revised manuscript. The probability distribution of measurements of $L_z$ is only stationary if $\frac{d}{dt}\langle L_z^n\rangle=0$ for all $n$. This is proven in Eq. (18). Also this proof is taken from a textbook. Any $\frac{d}{dt}\langle L_z^n\rangle\neq 0$ is a violation of $[H,L_z]=0$ and is experimentally detectable by measuring $L_z$, see section 2.5 and the new Fig. 1.

  6. Ground state mean-field solutions have been known to violate conservation laws since 1963 for Hartree-Fock and at least since 1975 for GP mean-field. Methods for restoring these symmetries have been textbook material since at least 1980. There is nothing ''misleading'' about stating that ''the mean field Gross-Pitaevskii (GP) description of a Bose-Einstein condensate (BEC) is unable to properly take into account angular momentum conservation''. It is a well-established fact. We only provide new results for the time-dependent case. See the newly rewritten introduction for details.

  7. We provide an explanation why satisfying conservation laws in the dynamics is only possible on the many-body level, see section 4.3 and 4.4, as well as parametric dependencies of the violations and how to fix the problem. None of this has been published before.

Referee 2 writes: ''However the fact the GP approach is unable in general to properly describe fluctuations is well known. It has nothing to do with BEC, but it is true for any mean-field approach. It would be rather surprising the opposite.''

Answer: Referee 2 is right that it is very well known that fluctuations are not accurately described by the GP mean-field or any other mean-field. However, fluctuations as such are not the topic of this work. The question we answer is what kind of approximations are capable of satisfying conservation laws such as angular and linear momentum conservation. It turns out that conservation laws can only be satisfied on the many-body level. This is not a priori clear. But we go far beyond this. Specifically, our findings are that angular momentum conservation is not only violated in the stationary case as has long been known, but that angular momentum conservation is violated by the GP time-dynamics. We provide quantitative predictions, including parametric dependencies for specific examples in a parameter regime, where one would expect the GP mean-field to provide very accurate predictions: the depletion of the condensate is less than $5\times 10^{-4}$. But most importantly we provide an explanation of this violation based on the variational principle. We even go further by gradually restoring the symmetry through extensive many-body simulations. None of the above has been published anywhere and it is very surprising that restoring this fundamental symmetry in the time-dynamics takes a tremendous computational effort. Please also see the new example we provide to demonstrate a violation of linear momentum conservation in section 8.

Referee 2 writes: ''2 - (given 1-) only the dipole mode with $L_z=0$ is considered. And only the second order $L_z^2$. Further cases are needed.''

Answer: 1. A single counter example is enough to prove a hypothesis wrong. We have provided such a counter example: GP theory violates angular momentum conservation. There is no need for further cases. Nevertheless, we have included another case, see the new section 8.

  1. We selected an analytically and numerically tractable case, to provide the parametric dependencies. This is why we chose the dipole mode.

  2. Referee 2 asks for another example. As is clear from the theory we provide in section 4, angular momentum is nothing special. We therefore decided to provide an additional example that shows that the GP mean-field also violates momentum conservation, $[H,P]=0$. Please see the new section 8.

  3. Referee 2 asks for higher moments. While no higher orders are needed to prove the violations we went the extra mile for referee 2 and evaluated $\langle L_z^3\rangle_{GP}$ and $\langle P^3\rangle_{GP}$ in Appendix B. As expected they are time-dependent as well.

Referee 2 writes: ''3 (given 1-) Comments on the use of Bogolyubov theory or linearised GP approach above the ground state in the linear regime to determine fluctuations - as done in literature - are completely absent.''

Answer: The referee is right that we did not use Bogolyubov theory. We have explained our findings based on the variational principle, i.e. the most fundamental level of explanation possible, see section 4. Unlike Bogolyubov theory or the linearized GP approach our approach does not rely on the assumption of a small depletion and linearization around a mean-field state. We use the full equations of motion. Obviously rigorous results are always preferable over approximate treatments when it is possible to obtain them. Thereby, we have excluded the possibility for loopholes. Had we only relied on linearized versions of the equations of motion, it could not be excluded that a violation of angular momentum conservation is merely a consequence of the linearization approximation.

Referee 2 writes: ''1. the fact that the error is not very large even in the dynamics is per se an interesting result (although pointing in the opposite direction of the author`s aim).''

Answer: 1. $\langle L_z\rangle=0$ is exactly constant in the dynamics. Even in the numerics at a level of the numerical precision $<10^{-8}$. In contrast, $\langle L_z^2\rangle$ varies over $\pm 13\%$, a difference of at least seven(!) orders of magnitude.

  1. As stated in the abstract, we chose very weak interaction strength and practically no depletion ($<5\times 10^{-4}$) to make the conditions as favorable as possible for the GP mean-field. Nevertheless, angular momentum conservation is heavily violated.

  2. Arbitrarily strong violations: $\frac{d}{dt}\langle L_z^2\rangle$ can take on any value between $-\infty$ and $\infty$ depending on the values chosen for $x_0, \sigma_0, p_0$, see Eq. (50). The violation grows linearly with the initial displacement from the center of the trap $x_0$, linearly with the initial momentum $p_0$, linearly with the GP nonlinearity parameter $\lambda$ and so on. It is no problem at all to find larger violations. To illustrate this fact we have included a new Fig. 2 to show these parametric dependencies.

  3. In section 8 we now provide an example for the violation of linear momentum conservation where $\langle P^2\rangle$ grows quickly by about 600% instead of staying constant.

  4. Note: many experiments work in the opposite limit where the kinetic energy is much smaller than the interaction energy. The violation grows with the interaction strength, see Eq. (50).

Conclusion: Referee 2 claims (incorrectly) that angular momentum is conserved by the GP mean-field dynamics, because $\frac{d}{dt}\langle L_z\rangle=0$. However, as we have shown, this is not enough in quantum mechanics: all moments $\langle L_z^n\rangle$ need to be time-independent. Mean-field violations of conservation laws have been known for many decades for stationary states. Here, we treat the time-dependent case. We explain from first principles why conservation laws can only be satisfied on the many-body level in the dynamics. In order to avoid further confusion we have included all necessary background material that illustrates these points. Following referee 2's requests we included another example, showing the violation of momentum conservation by the GP mean-field and provide results for higher order momenta.

---

## Round 2 · Referee Report · Anonymous · 2019-2-5

Strengths
The paper improved in the resubmitted version, and in particular:
1 - an historic discussion and perspective has been added, clarifying the point raised by the authors;
2 - a discussion of the momentum conservation has been added, again clarifying the point raised by the authors.
Weaknesses
In my opinion, the main weakness of the previous version remain unaltered, actually it further increased: the authors put all the emphasis on the violation of the conservation laws by performing mean-field approximations, while it is well known that such approximations do not reproduce correlation functions and higher momenta.
Report
I read with interest both the revised version, and the detailed reply of the authors. To me, the main point of interest of the paper is the quantification of the errors committed by performing the mean-field, both at equilibrium and during the dynamics. However, in the present version the emphasis is almost entirely given to the need of inserting the many-body theory to restore the symmetry. I think the point is far from being unexpected: consider the Ising model in a transverse field (in one or more dimensions, so independently from the fact that is integrable/solvable or not), and then perform mean-field. Everybody would agree that the values for the ground-state energy and the other observables are affected by some error, that errors are committed during the dynamics, and that the higher momenta or the correlations are not well reproduced. The interesting point is to quantify such deviations, and to demonstrate that by a series of controllable approximation one can reduce them - but not that it is needed the full theory to correctly conserve all higher-order expectation values. So the paper is convincing in points like figure 6, where the convergence is studied - not convincing when it is showing in several ways something expected such that there is a failure in the determination of expectation values of observables like A^n.
Finally, despite I agree that mean-field approaches does not reproduce higher correlation functions, a simple example can show that in mean-field expectation values of higher power of observables are conserved. The example is the following: suppose that one solve the time-independent GPE as
H_GPE \psi_n(r)=\mu_n \psi_n(r). It is obvious that choosing as initial condition \psi(r,t=0)=\psi_n(r), then \psi(r,t)=e^{-i \mu_n t/\hbar} \psi_n(r). Compute now the mean-field expectation value of p^{2n}, defined <p^{2n}>_MF=\int dr \psi(r,t)*\ast p^{2m} p^{2n} \psi(r,t). It is clear that <p^{2n}>_MF does not depend on time. No contradiction with the previous statements is present, since <...>_MF is different from <...>_TRUE. This simple example confirm that what is truly interesting, and partly done in this paper, is to quantify the deviation of expectation values in mean-field from the exact ones.
So, from one side it is completely on the author'side to choose the emphasis they want to give. However, from my referee's side, I conclude that with a misdirected emphasis, the readability and the quality of the paper is affected, so that I cannot suggest publication of the paper in the present version.

---

## Round 2 · Referee Report · Anonymous · 2019-2-8

Strengths
1- The results on how much the GP fails in describing the dynamics some higher momenta of L_z and now also for P, and the possibility, with not that many orbitals, to heal at least partially the problem (but only for L_z).
2 - The new version - with the added material -better explains the point of the authors.
3 - In the new version the readability has been improved and it is written in a pedagogical way.
Weaknesses
As for the original version the weakness of the manuscript resides in the claim, in the emphasis the authors put on their results, namely that it is surprising that the GP mean-field equation does not conserve the symmetries of the many-body system.
As already pointed out in previous referee's reports and comments the fact that mean-field approaches fail in describing correlations - and therefore the particular instance considered by the authors of higher momenta of observable - is well known.
The weakness is even reinforced in the new version. The authors instead of amending their claim, they put even more emphasis on it.
Report
In the first round the referees have pointed out that it is well known that mean field approaches are not expected to properly describe correlations (also dynamically) and in particular higher momenta of observables. Therefore in general the symmetries of the microscopic Hamiltonian are not satisfied.
Although the authors have not appreciated it, both referees mentioned that the failure of the GPE of properly describing the evolution of <L_z> would have been very surprisingly. Not, however, the failure in describing higher momenta of L_z or any other operator.
In the new version the authors instead of amending the original claim, put all the emphasis - also in the reply to the referee's reports - on explaining that many body theory is needed to preserve/recover the symmetries of the microscopic Hamiltonian.
In this respect let me also notice a small logical loophole in the reply of the authors.
As the author mention it is well known (appendix A is rather useless indeed) that conservation laws related to a generator A implies the d/dt<A^n>=0 for any n. If the latter is not satisfied the global symmetry is not preserved by the approximation used. The author in particular use the fact that for n=2 GP shows some evolution and they claim this effect being surprising and therefore the need to go beyond it.
On the other hand they also write in the reply that:
" [..] it is very well known that fluctuations are not accurately described by the GP mean-field or any other mean-field. However, fluctuations as such are not the topic of this work."
Now fluctuations correspond to the case n=2 and therefore it is also not surprising that GP is not symmetry conserving.
As for the previous round I cannot recommend the publication of the manuscript in its present form,
since the authors' aim is to convey the massage that they have discover that GP mean-field equations do not properly describe higher momenta of observables (and therefore for the observables generators of a symmetry of the Hamiltonian, GP does not preserve the symmetry of it). What it is interesting is to quantify the failure of GP for some quantities and how "hard" is to cure it.
* * *
Concerning the (added) results:
1. The results on <P^2> and the emphasis (in the reply) put by the authors on the deviation of the GP result even when the depletion is negligible, is almost not new.
It has being discuss in the context of uncertainty in ref [38], where the authors offer also an explanation of why the large deviation in the time evolution occurs within a GP approach.
2. Sec 2.5 should be an Appendix if they want to keep it.
3. Appendix A is textbook.
As a final comment let me stress that I do recognize clear merits of the manuscript: it is well written (forgetting the parts concerning their major claim) and with a pedagogical aim; they use some quite general examples to quantify, at least to some extent, the failure of the GP in describing the evolution of n=2 (and n=3) momenta and show how including more orbitals the situation is improved.
I find their results together with the ones of Ref [38] and of the not-cited paper by Klaiman and Cederbaum [PHYSICAL REVIEW A
94, 063648 (2016)] (which in my opinion has to be cited) useful to get a better understanding on the GP mean-field equations.

---

## Round 2 · Author Response

List of changes
* introduction: discussion of the history of mean-field violations of conservation laws since 1963
* section 2: instead of specializing to just angular momentum we have generalized the theoretical treatment to arbitrary conservation laws
* a new section 2.4: discussion of the full distribution function and its relation to the moment problem
* a new section 2.5: discussion of a Gedankenexperiment using a single particle to show explicitly that d/dt<A> is not a sufficient condition for the conservation of an observable A. Discussion of the inconsistencies that arise if [H,A]=0 is replaced by the weaker condition d/dt<A>=0.
* a new Fig. 1 to illustrate the Gedankenexperiment in section 2.5 for a specific example.
* a new Figure 2 that demonstrates that arbitrarily large violations of angular momentum conservation are possible in GP mean-field dynamics at fixed interaction strength depending on the parameters of the initial state.
* a new section 8 discussing the violation of linear momentum conservation by the GP mean-field dynamics using stronger interaction, including analytical calculations, analogous to the case of angular momentum.
* a new Figure 7 demonstrating the violation of linear momentum conservation by the GP mean-field in a numerical example (a violation of ~600%).
* a new Appendix A containing an alternative proof for the fact that conserved observables have stationary moments, d/dt<A^n>=0 for all n. This proof is taken from the textbook "Quantum Mechanics" by Cohen-Tannoudji, Diu and Laloe.
* a new Appendix B containing analytical results for the third moments d/dt<L_z^3> and d/dt<P^3> in GP mean-field. These are nonzero, just like the second moments.
* a new Appendix C demonstrating that kinietic energy is not conserved for an interacting system even though d/dt<P^2>=0 in the exact dynamics.

You are currently on this page

---

## Round 2 · List of Changes

* introduction: discussion of the history of mean-field violations of conservation laws since 1963
* section 2: instead of specializing to just angular momentum we have generalized the theoretical treatment to arbitrary conservation laws
* a new section 2.4: discussion of the full distribution function and its relation to the moment problem
* a new section 2.5: discussion of a Gedankenexperiment using a single particle to show explicitly that d/dt<A> is not a sufficient condition for the conservation of an observable A. Discussion of the inconsistencies that arise if [H,A]=0 is replaced by the weaker condition d/dt<A>=0.
* a new Fig. 1 to illustrate the Gedankenexperiment in section 2.5 for a specific example.
* a new Figure 2 that demonstrates that arbitrarily large violations of angular momentum conservation are possible in GP mean-field dynamics at fixed interaction strength depending on the parameters of the initial state.
* a new section 8 discussing the violation of linear momentum conservation by the GP mean-field dynamics using stronger interaction, including analytical calculations, analogous to the case of angular momentum.
* a new Figure 7 demonstrating the violation of linear momentum conservation by the GP mean-field in a numerical example (a violation of ~600%).
* a new Appendix A containing an alternative proof for the fact that conserved observables have stationary moments, d/dt<A^n>=0 for all n. This proof is taken from the textbook "Quantum Mechanics" by Cohen-Tannoudji, Diu and Laloe.
* a new Appendix B containing analytical results for the third moments d/dt<L_z^3> and d/dt<P^3> in GP mean-field. These are nonzero, just like the second moments.
* a new Appendix C demonstrating that kinietic energy is not conserved for an interacting system even though d/dt<P^2>=0 in the exact dynamics.

You are currently on this page

---

## Editorial Decision

unknown